# Thermal radiation control from hot graphene electrons coupled to a photonic crystal nanocavity

Ren-Jye Shiue[1], Yuanda Gao[2], Cheng Tan[2,3], Cheng Peng[1], Jiabao Zheng[1,3], Dmitri K. Efetov[4], Young Duck Kim [2,5], James Hone[2] & Dirk Englund [1]

Controlling thermal radiation is central in a range of applications including sensing, energy harvesting, and lighting. The thermal emission spectrum can be strongly modified through the electromagnetic local density of states (EM LDOS) in nanoscale-patterned metals and semiconductors. However, these materials become unstable at high temperature, preventing improvements in radiative efficiency and applications such as thermophotovoltaics. Here, we report stable high-temperature thermal emission based on hot electrons (>2000 K) in graphene coupled to a photonic crystal nanocavity, which strongly modifies the EM LDOS. The electron bath in graphene is highly decoupled from lattice phonons, allowing a comparatively cool temperature (700 K) of the photonic crystal nanocavity. This thermal decoupling of hot electrons from the LDOS-engineered substrate opens a broad design space for thermal emission control that would be challenging or impossible with heated nanoscale-patterned metals or semiconductor materials.

[1] Department of Electrical Engineering and Computer Science, Massachusetts Institute of Technology, Cambridge, MA 02139, USA. [2] Department of Mechanical Engineering, Columbia University, New York, NY 10027, USA. [3] Department of Electrical Engineering, Columbia University, New York, NY 10027, USA. [4] ICFO-Institut de Ciencies Fotoniques, The Barcelona Institute of Science and Technology, 08860 Castelldefels, Barcelona, Spain. [5] Department of Physics, Kyung Hee University, Seoul 02447, Republic of Korea. These authors contributed equally: Ren-Jye Shiue, Yuanda Gao. Correspondence and requests for materials should be addressed to D.E. (email: englund@mit.edu)

Thermal radiation of a blackbody generally exhibits a broadband spectrum that depends on the emissivity and temperature of the thermal emitter, as described by Planck's law. The desire to control the thermal emission spectrum has a long history, notably including the 1885 invention by Carl Auer von Welsbach of the Actinophor gas mantle, which dramatically improved gas-lamp radiative efficiency, and in modern times the development of spectrally selective thermal emitters to boost the efficiencies of solar energy harvesting[1,2] and illumination[3].

In the modern formulation of blackbody radiation[4], the spectral energy density of a blackbody follows $u(\omega, T) = E(\omega)n(\omega, T) D(\omega)$, where $E(\omega)$ and $n(\omega, T)$ are the mode energy and the mode photon occupation, respectively, and $D(\omega)$ is the electromagnetic local density of states (EM LDOS). By controlling the LDOS in a sub-wavelength optical structure, it is possible to strongly modify the thermal emission spectrum. A variety of structures have been developed to tailor thermal radiation in this way, including optical gratings[5], photonic crystals[1,6,7], photonic cavities[8,9], nano-antenna[10], and metamaterials[11–13]. These demonstrations highlight the control of thermal emission by control of the LDOS, but face challenges in high-temperature stability as melting, evaporation, chemical reactions, surface diffusion, and delamination become severe for these nanoscale-patterned metallic and semiconducting materials. Thus, although high-temperature thermal emitters could greatly increase radiative efficiency for high-temperature thermophotovoltaics (TPVs) and light sources, it remains difficult to tailor thermal emission at elevated temperatures beyond about 1000 K by engineering LDOS.

Nanocarbon-based materials, including graphene[14–19] and carbon nanotubes[20–22], have emerged as intriguing thermal emitters alternative to metal and semiconductor-based materials. Previous demonstrations show that they can support high saturation current density[14,17,19], ultrafast heating (cooling) modulation[15,18,22], and flexible integration with existing electronic and photonic technology[16,20,21]. Here, the graphene–photonic crystal structure thermal emitter system addresses the challenges of high-temperature thermal radiation control by (i) direct heating of the electron gas in monolayer graphene, and (ii) coupling of this thermal emitter to a silicon planar photonic crystal (PPC) nanocavity. This approach has two key advantages: (i) the thermal emission that arises from the graphene electron gas, whose temperature is highly decoupled from graphene's atomic lattice, can exceed 2000 K, while the surrounding Si cavity itself stays at only 700 K. (ii) The PPC cavity strongly modifies the LDOS, producing a sharp redistribution of the hot electrons' thermal emission into the desired spectral regions.

## Results

### Device design.
As shown in Fig. 1, the device consists of a graphene/hexagonal boron-nitride (hBN) heterostructure on top of a Si PPC cavity. We use Joule heating by an electric current through the graphene sheet to raise the electron gas temperature to produce thermal radiation similar to a heated gray body[14]. Because hot electrons in graphene thermalize much faster via electron–electron and optical phonon scattering than acoustic phonon scattering[23–26], the heated electrons first reach equilibrium with optical phonons in graphene and hBN, coupling more slowly to the acoustic phonon bath by thermal conductance $\gamma_e$. The heat eventually dissipates to the silicon substrate (via $\gamma_0$), which can remain at a much lower temperature than the electron gas of graphene, as depicted in Fig. 1c. As will be shown, the electron gas in the graphene monolayer can reach very high temperature exceeding 2000 K, resulting in strong thermal radiation in the infrared and visible spectra.

We created the wavelength-scale PPC cavity by introducing a line defect with shifted air holes near the center[27]. An air-slot along the center of the PPC confines light longitudinally and extends the optical modes above and below the silicon membrane[27,28]. Finite-difference time-domain (FDTD) simulations indicate strongly confined optical modes in the air-slot cavity, as seen in the $|\mathbf{E}|^2$ in Fig. 1d, where $\mathbf{E}$ is the electric field. This air-slot cavity increases the coupling rate with a 2D material on the PPC surface by almost a factor of three compared with a linear three-hole defect (L3) cavity[27,29]. We further etched the hBN/graphene/hBN stack into a bowtie shape and aligned the central narrow strip to the PPC cavity area. This bowtie-shaped graphene facilitates a heated hot-electron spot in the graphene device to achieve optimal coupling of the hot-electron radiation to the cavity resonant field. Figure 1b shows the optical image of the finished device.

We characterized the PPC cavity using a cross-polarized confocal microscope with a broadband excitation source (a supercontinuum laser) vertically coupled to the cavity. The reflection spectrum of the cavity before graphene deposition (Fig. 1e, blue curve) indicates three narrow resonances at 1488.9, 1496, and 1511.2 nm with quality factors $Q$ of 1500, 2000, and 2300, respectively. After the hBN/Graphene/hBN deposition, the cavity resonances red-shifted to 1559.1, 1568.2, and 1590.7 nm, respectively, due to higher refractive index of hBN and graphene than air. The $Q$ factors dropped to 520, 430, and 400, respectively, because of the excess absorption of graphene to the cavity field. The degradation of $Q$ due to the 25-nm-thick hBN layers is negligible, as tested in separate PPC cavities without graphene, consistent with simulations (See Supplementary Note 4).

### Cavity-graphene thermal emission.
Figure 2a plots the drain-source current $I_{DS}$ (blue) of the graphene device as a function of the applied drain-source voltage $V_{DS}$. The differential resistance $R_{diff} = (dI_{DS}/dV_{DS})^{-1}$ increases with $V_{DS}$, which is a signature of self-heating and strong electron scattering by hot optical phonons in graphene and hBN[30–33]. The measured thermal emission spectra, plotted in Fig. 2b for $V_{DS}$ voltages from 10 to 13 V, show three pronounced narrowband peaks that match the the cavity resonant modes as obtained from the reflection measurement shown in Fig. 1e.

We can extract the temperature of the hot graphene electrons from thermal emission spectra and the absorption of graphene in the cavity. From Kirchoff's law, the emissivity of graphene is equal to its absorption, which can be obtained from temporal coupled-mode theory that incorporates the coupling of graphene to the optical modes inside a cavity[34]. The frequency-dependent absorption of graphene can the be expressed as

$$A_g(\omega) = \frac{\frac{1}{Q_0}\left(1/Q_g - 1/Q_0\right)}{\left(1 - \omega/\omega_0\right)^2 + \left(1/2Q_g\right)^2} = \epsilon_g(\omega) \qquad (1)$$

where $Q_0$ and $Q_g$ are the quality factors of the cavity before and after graphene deposition, respectively, $\omega_0$ is the resonant frequency and $\epsilon_g(\omega)$ is the emissivity of graphene. In our spectroscopy setup, the radiation of the cavity-graphene only couples to the microscope objective mode with a coupling efficiency $\eta \sim 0.1$, as calculated from 3D FDTD simulations. The spectral radiance of the cavity-graphene emitter therefore equals $I(\omega) = \eta\epsilon_g(\omega)I_{BB}(\omega) = 0.072L(\omega)I_{BB}(\omega)$, where $I_{BB}(\omega) = \frac{\hbar\omega^3}{4\pi^3c^2}\left(e^{\hbar\omega/kT_{BB}} - 1\right)^{-1}$ is the spectral radiance of an ideal blackbody, $T_{BB}$ is the blackbody temperature, and $L(\omega) = \epsilon_g(\omega)/\epsilon_g(\omega_0)$ denotes the normalized Lorentz spectrum of the

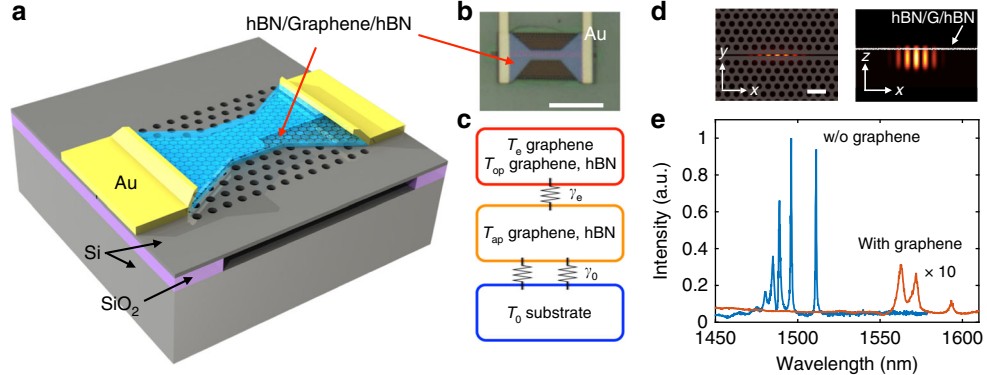

**Fig. 1** Overview of the cavity-graphene thermal emitter. **a** Schematic of a cavity-integrated hBN/graphene/hBN light emitter with edge contacts. **b** Optical image of the fabricated device. Scale bar: 5 μ*m*. **c** Schematic of energy relaxation of graphene hot electrons. The red block corresponds to quasi-equilibrium of hot graphene electrons and optical phonons of graphene and hBN. Subsequently, the heat flows to the acoustic phonons and the substrate. **d** FDTD simulation of the electric field profile $|\mathbf{E}|^2$ of the silicon PPC air-slot cavity indicates strongly confined resonant modes. Scale bar: 1 μm. **e** Reflection spectra of the PPC cavity before (blue) and after (red) deposition of graphene on the PPC surface

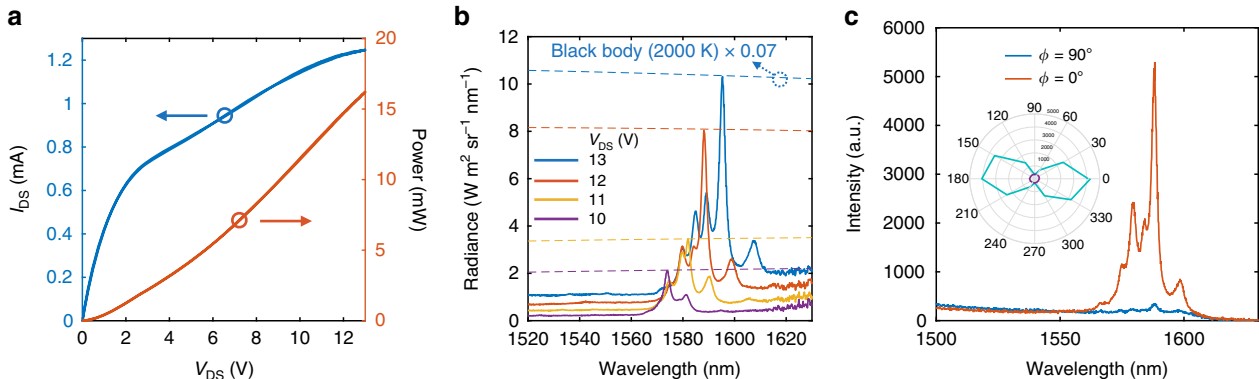

**Fig. 2** Thermal radiation properties. **a** Current–voltage (blue) curve of the hBN/graphene/hBN emitter. The red curve corresponds to the electrical power that is applied to the graphene emitter. **b** Emission spectra of the graphene emitter at different $V_{DS}$ voltages. The dashed reference blackbody emission spectra at 2000 K (blue), 1930 K (red), 1640 K (yellow), 1510 K (purple) serves to compare with the emission of the graphene emitter. **c** The emission spectra of the graphene emitter at polarization angles of $\phi = 0°$ (red) and $\phi = 90°$ (blue) with applied $V_{DS} = 13$ V. Inset shows the emission intensity with respect to $\phi$ at wavelengths of 1585 nm (cyan) and 1500 nm (purple)

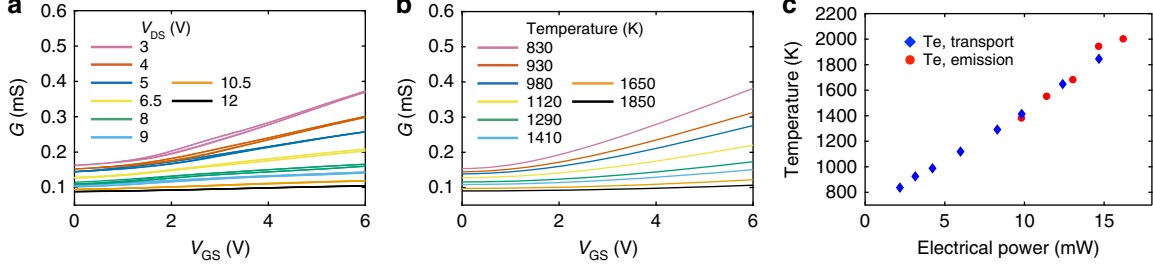

**Fig. 3** Electrical transport of hot graphene electrons. **a** Gate-dependent conductance of the graphene device measured at different $V_{DS}$ voltages. The elevated electronic temperature of graphene due to applied $V_{DS}$ voltages results in the weaker variation of the conductance via electrostatic gating. **b** Theoretical calculation of the graphene conductance with different electronic temperatures. **c** The electron temperature of the graphene extracted from thermal emission spectra (red circles) and electrical transport model (blue diamonds)

cavity. The blue dashed line in Fig. 2b shows a calibrated blackbody spectrum at $T_{BB} = 2000$ K, matching the measured peak intensity of the cavity-graphene emitter at $V_{DS} = 13$ V. From the radiation peak intensity $I(\omega_0)$ of the graphene emitter at different bias voltages ($V_{DS}$), it is possible to extract the electron temperatures of graphene with respect to $V_{DS}$, as the red circles shown in Fig. 3c.

Figure 2c shows the polarization-dependent intensity of the thermal emitter with $V_{DS} = 13$ V. The red and blue curves represent the emission spectra that are collected at polarization angles parallel ($\phi = 0°$) and perpendicular ($\phi = 90°$) to the *x*-axis of the cavity. The intensity of the thermal emission collected at $\phi = 0°$ shows a strong, 15-fold enhancement compared with that collected at $\phi = 90°$ at the resonant wavelength of 1585 nm. The polar plot in the inset of Fig. 2c shows the emission intensity with respect to the angle $\phi$ at the resonant (cyan) and non-resonant (purple) wavelengths. A clear polarization-dependent on-resonant emission spectrum that varies

with cos $\phi$ is consistent with the radiation of the cavity resonant mode and further confirms the coupling of the thermal emission of graphene to the PPC cavity modes.

**Electrical transport of hot graphene electrons**. The elevation of the electron temperature ($T_e$) in graphene also affects graphene's electronic transport properties, providing a second approach to validate $T_e$ in graphene. As shown in Fig. 1a, the silicon membrane serves as a global gate to electrostatically tune the carrier density in graphene. Figure 3a shows the conductance, $G$, of the graphene sheet as a function of gate-source voltage ($V_{GS}$) at different $V_{DS}$ voltages. When $V_{DS}$ is small, the thermally activated free carrier density $n_{th}$ in graphene is less than the gate-source voltage controlled carrier density $n_g$. Thus, $n_g$ dominates the conduction of graphene, and $G$ shows a strong gate ($V_{GS}$)-dependence. At larger $V_{DS}$, the elevated electron temperature $T_e$ results in higher $n_{th}$ that then dominates the conductance of graphene. Therefore, $G$ only shows weak variation as $V_{GS}$ ($n_g$) changes.

A simple model quantitatively explains these electrical measurements and allows us to extract the graphene temperature, which can then be compared with independent optical measurements. From the literature[35,36], we have

$$n_{th} = \frac{\pi}{6}\left(\frac{k_B T_e}{\hbar v_F}\right)^2 \left(1 + e^{-(T_e/T_0-1)/2}\sqrt{T_e/T_0 - 1}\right) \quad (2)$$

$$n_g = C_{ox}(V_{GS} - V_D)/e \quad (3)$$

$$n_e(n_h) = \frac{1}{2}\left(\pm n_g + \sqrt{n_g^2 + 4n_{th}^2 + n_{pd}^2}\right), \quad (4)$$

where $n_{pd}$ is the carrier density due to electron–hole puddles in graphene, $C_{ox}$ is the gate capacitance, $T_0$ is the ambient temperature, and $V_D$, $v_F$ are the charge neutrality voltage and the Fermi velocity of graphene, respectively. The conductivity of graphene is given by $\sigma = (n_h + n_e)e\mu(T_e)$, where $e$ is the electron charge. In our model, we use a temperature-dependent mobility of graphene based on a drift velocity-field relation[35,37,38], giving $\mu(T_e) = \mu_0(T_0/T_e)^\beta$, where $\mu_0 = 20{,}000$ cm$^2$ V$^{-1}$ s$^{-1}$ is the mobility of graphene at 300 K and $\beta = 2.3$ is extracted from the electro-thermal simulation based on the measured current–voltage in Fig. 2a (see Supplementary Note 1). Due to the bowtie shape of the graphene emitter, the integral of the total conductance $G$ along the source-drain channel is $G = \frac{\sigma}{\xi} // \frac{1}{r_c}$, where $\xi$ ~4 is the geometry factor and $r_c$ is the temperature-dependent contact resistance of graphene (Supplementary Note 1). Fitting the curves in Fig. 3a to the above model provides the electrical conductance curves shown in Fig. 3b. The extracted $T_e$ at different $V_{DS}$ voltages

are shown in Fig. 3c. The calculated temperature from the electrical modeling agrees well with temperature deduced from the emission at $V_{DS} = 10$ to 13 V.

**PPC cavity temperature**. The thermal emission in Fig. 2b shows a red-shifting of the cavity with increased current. This resonance shift is primarily due to the thermo-optic effect of Si and thus allows us to extract the silicon PPC cavity temperature. In Fig. 4a, the blue and yellow curves show two examples of reflection spectra of the PPC cavity under $V_{DS} = 2$ and 4 V of graphene, respectively. Figure 4b shows the cavity wavelength shift (red curve) as a function of $V_{DS}$. Using FDTD simulations, we deduce the cavity temperature from the cavity resonance redshift, based on a thermo-optic coefficient[39] of silicon of $1 \times 10^{-4}$ K$^{-1}$. The wavelength shift is proportional to the silicon cavity temperature ($T_{Si}$) with $\lambda(T_{Si}) = \lambda(T_0) + \alpha(T_{Si} - T_0)$, where $\alpha = 0.1$ nm·K$^{-1}$ and $T_0$ is the ambient temperature (300 K). Figure 4b plots the extracted temperature (blue curve) of the cavity with respect to $V_{DS}$.

A striking conclusion is that the electron gas temperature of graphene (Fig. 3c) far exceeds the suspended silicon PPC cavity temperature (Fig. 4b): the electron gas can be hot—as desired for thermal radiator—while the nanophotonic substrate that strongly modifies the emission spectrum through the EM LDOS, remains comparatively cool. These results show that the unusually low coupling between the graphene electron gas and graphene acoustic phonons allows a hot thermal emitter to be coupled to a nanostructured optical medium whose tailored EM LDOS sharply modifies the emission spectrum. To our knowledge, this work represents the strongest modification of black-body radiation in the near-infrared spectrum.

**Temporal response of graphene emission**. As shown in Fig. 4c, we investigated the temporal response of the graphene emission with an 100-ps electrical pulse excitation (see Methods). The temporal response of the thermal emission shows a full width at half maximum (FWHM) of 350 ps, indicating an on–off modulation speed >1 GHz. Varying the excitation pulse duration $\Delta T$ from 0.1 to 2 ns, we observed that the emission intensity started to saturate for $\Delta T > 1$ ns, corresponding to a saturation temperature of 1550 K, as shown in Supplementary Fig. 4. We numerically simulated the transient temperature of the graphene emitter, showing that the substrate temperature elevation can be reduced to only 60 K with 10%-duty-cycle electrical pulses (See Supplementary Note 3). As we demonstrated in separate work[15], an optimized graphene device allows an on–off modulation speed of the thermal emission at a rate exceeding 10 GHz—comparable to fast gain-switched lasers.

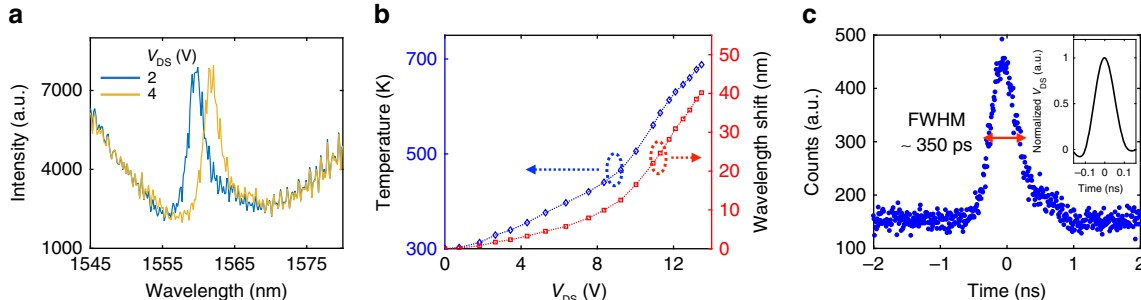

**Fig. 4** PPC cavity temperature and time-dependent thermal emission. **a** The reflection spectra of the PPC cavity at applied $V_{DS}$ voltages 2 and 4 V on the graphene emitter. The cavity resonance shows constant red-shifting as $V_{DS}$ increases. **b** The wavelength shift (red) and the extracted temperature (blue) of the PPC cavity with respect to $V_{DS}$. **c** Generation of a short (350 ps) thermal radiation pulses from the cavity-graphene emitter by applying an 100-ps electrical pulse. Inset: temporal profile of the electrical pulse

## Discussion

This chip-integrated, spectrally controlled black-body radiator can serve as a useful light source for optical communications with low-power requirements. For example, many forms of discrete-variable quantum key distribution (QKD) require light sources that can be modulated at several GHz, in which consecutive pulses are phase-randomized (which is automatically provided here by the thermal emission), and with <1 photon per pulse[40]. We estimate that the cavity-coupled hot-graphene light source demonstrated here would inject 0.2 photons into a waveguide per 100-ps pulse (See Supplementary Note 5). Thus, this light source should be suitable for discrete-variable QKD applications. Another application is in the recently developed protocol of "Floodlight QKD", which requires a broadband light source on the transmitter with fewer than one photon per spatial-temporal mode[41]. We estimate that the source can also be of use as a thermal light source for on-chip spectroscopy, as well as related sensing applications. In the application of TPV, Ohmic heating would be replaced by sunlight absorption into the graphene sheet. In this application, it may be necessary to replace the graphene monolayer by a multilayer stack to provide higher absorption into the electron gas, but we expect the device to work similarly otherwise, since the high-energy optical phonon (>0.15 eV) still thermalize much faster than acoustic phonons in multilayer graphene and graphite[42–44].

We briefly comment on the key features that lead to the high modulation speed and the exceptionally high temperature of the thermal emitter. (1) The bowtie shape of the hBN-encapsulated graphene allows precise spatial coupling of the hot electron gas at its narrowest section with the nanocavity; future work could further improve the mode overlap by a smaller (<100 nm) taper of the graphene layer coupled to slot-mode nanocavities[28,45]. (2) The fan-out of the graphene layer to the metal contacts produces relatively low contact resistance, approximated to be 65 Ω per contact (See Supplementary Note 6); this resistance may be further reduced by graphite/graphene contact with optimal orientation[46]. The S11 in Supplementary Fig. 7b indicates an RC-limited response time ~0.19 ns (See Supplementary Note 7). This response time is also similar to the expected acoustic phonon cooling time ~0.2 ns. (3) The high electron temperature reaching up to 2000 K was enabled in part by the weak coupling between graphene electrons and acoustic phonons and the hBN encapsulation[14,15,19]; for comparison, previous demonstration with CNTs and graphene without hBN encapsulation reported 1500 K[21] and 1100 K[16], respectively. A temperature increase from 1500 to 2000 K translates to a threefold higher thermal radiation intensity since the thermal radiated power scales as $T^4$ (Stefan–Boltzmann law). Even higher electron gas temperatures up to 2800 K, which were reported for suspended graphene monolayers, could further improve the emitter's radiative efficiency.

In conclusion, we have demonstrated an electrically-driven on-chip thermal emitter based on a hBN/graphene heterostructure. The hBN-encapsulated graphene device achieves a temperature up to 2000 K through Joule heating, producing pronounced thermal radiation in the infrared spectrum. Coupling this hot-electron thermal emitter to a PPC cavity enables spectrally-selective thermal radiation control at a stable emission temperature of 2000 K, while the cavity itself remains cool (700 K) because of mismatched electron-optical photon and electron-acoustic phonon coupling in graphene. This device enabled the strongest modification of the black-body radiation in the near-infrared spectrum. Because of the broadband absorption of graphene and its stable emission temperature of more than 2000 K, this graphene-PPC cavity concept can be extended to shorter (into the visible spectrum[14]) and longer wavelength (mid-IR and beyond) spectral regions.

Two-dimensional materials and their heterostructures have shown great flexibility for their assembly onto a variety of bulk materials and their photonic systems. These heterogeneously integrated 2D photonic components have emerged as a versatile platform for photodetectors[47], electro-optic modulators[29], light emitting diodes (LEDs)[48,49], plasmonic[50], and nonlinear optical devices[51,52]. In separate experiments, we have shown that the thermal emitter can be on–off modulated at rates exceeding 10 GHz[15], suggesting applications for easy-to-integrate on-chip optical interconnect or QKD light sources. The tailored narrow-band thermal emission spectrum and the flexible integration of 2D materials may also find applications in thermophotovoltaics, and on-chip light sources for sensing[53,54] and spectroscopy.

## Methods

**Device fabrication.** The PPC cavities were fabricated on a silicon-on-insulator (SOI) wafer using a series of electron-beam lithography (EBL), reactive ion etching, and a wet-etch undercut of the insulator to produce free-standing membranes. The silicon membrane has a thickness of 220 nm with a PPC lattice period of $a = 470$ nm and an air hole radius $r = 0.29a$. A 30-nm-thick hafnium oxide (HfO₂) layer deposited on the PPC substrate by atomic layer deposition ensures electrical isolation between Si and the 2D materials.

Graphene and hBN were prepared by mechanical exfoliation. We then transferred the exfoliated hBN/graphene/hBN stack onto the PPC using a van der Waals assembly technique[55]. The total thickness of the two BN layers is around 25 nm. Patterning the hBN/Graphene/hBN stack with hydrogen-silsesquioxane (HSQ) resist and CHF₃ + O₂ plasma exposed the edges of graphene, which were subsequently contacted by Cr/Pd/Au (1/20/50 nm) metal leads using electron-beam evaporation. The entire device is then etched again by CHF₃ + O₂ plasma to form the bowtie geometry.

**Spectroscopy apparatus.** Supplementary Figure 2 shows the schematic of the optical measurement setup. The sample was mounted on a 3-axis translational stage under an objective with a numerical aperture (NA) of 0.55. The input supercontinuum laser source was first coupled to a linear polarizer (LP) and a beam splitter, then exciting the sample from vertical incidence. The output light was collected at a polarization angle perpendicular to the first LP, being analyzed by a spectrometer (Princeton Instruments SP2500). The spectrometer consists of a grating of 300 grooves/mm centered at 1.2 μm wavelengths and a liquid-nitrogen-cooled InGaAs detector array for recording light intensity. The same system serves to characterize the thermal radiation spectrum.

The throughput of the optical system was calibrated using a calibrated blackbody (BB) source from 373 to 1255 K with an emissivity of 0.99 (OMEGA-BB-4A). For temperature >1255 K, we first normalized the detector response to the BB source measured at moderate temperature (1255 K), and then verified the linear response of the InGaAs detector using the supercontinuum laser at high optical intensity to ensure a valid calibration. Based on the calibrated system response, the blackbody spectral radiance can be obtained considering the collection of a Gaussian spatial mode by the objective with surface area $\pi W_0^2$ and solid angle $\Delta \Omega = \lambda^2 / \pi W_0^2$, where $\lambda$ is the radiation wavelengths and $W_0$ is the Gaussian beam waist size. The blue dashed line in Supplementary Fig. 3 displays a calibrated BB spectrum at $T_{BB} = 2000$ K scaled by 0.072 to compare with the measured cavity-graphene radiation. Figure 3b shows the emission spectra after conversion based on the calibrated BB data.

**Time-resolved thermal emission measurement.** We used a time-correlated single photon counting (TCSPC) method to gauge the temporal response of the cavity-graphene emission in Fig. 4c. Electrical pulses generated by a pulse pattern generator (Anritsu MP1763B) and DC voltages from a source meter (Keithley K2400) were first coupled to a bias tee (MITEQ). The electrical pulse has a peak-to-peak voltage of 2 V and the DC voltage is 7 V. The mixed electrical signal was then coupled to the source-drain of the graphene device to modulate the thermal emission intensity. The generated emission photons from graphene was collected to a Si single-photon detector with timing resolution ~150 ps (PicoQuant); the photon incident events were recorded by the photon counting electronics (PicoQuant PicoHarp 300), which was synchronized by the same pattern generator (Anritsu MP1763B) with 100 MHz repetition rate. The applied electrical pulse waveform for the TCSPC measurement is displayed in the inset of Fig. 4c. The same technique is applied to measure the time-resolved thermal emission in Supplementary Note 2 with a DC voltage of 9 V and electrical pulse durations varying from 0.1 to 2.0 ns.

**Numerical simulations.** The cavity mode profile and far-field emission pattern shown in Fig. 1d and Supplementary Fig. 6 were simulated using a 3D FDTD simulation method (Lumerical FDTD Solutions). The simulation grid was $a/60$, and the refractive index of silicon and hBN were 3.47 and 1.8, respectively. The electro-thermal simulation shown in Supplementary Fig. 1 were performed using a finite element method in COMSOL Multiphysics with AC/DC and heat transfer modules.

## Data availability

The data that support the findings of this study are available from the corresponding author upon request.

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

## Acknowledgements

The authors thank Hyeongrak Choi for insightful comments and discussions. Y.G. and J. H. acknowledge support from the US Office of Naval Research N00014-13-1-0662. C.P. was supported in part by the Army Research Office (grant 16112776) and in part by the Stata Family Presidential Fellowship of MIT. D.E. acknowledges support from the U.S. Army Research Office through the Institute for Soldier Nanotechnologies. This work is supported in part by the Semiconductor Research Corporation's NRI Center for Institute for Nanoelectronics Discovery and Exploration (INDEX). The device fabrication was carried out in part at the Center for Functional Nanomaterials, Brookhaven National Laboratory, which is supported by the U.S. Department of Energy, Office of Basic Energy Sciences, under Contract No. DE-SC0012704. R.-J.S. is supported in part by Centers for Excitonics, and Energy Frontier Research Center funded by Department of Energy, Office of Science, Office of Basic Energy Sciences under award no. DE-SC0001088.

## Author contributions

R.J.S., Y.G., D.E., and J.H. conceived the experiment. Y.G., R.J.S., C.T., and J.Z. fabricated the samples. R.J.S., D.K.E., C.P., and Y.G. constructed the measuring setup and performed the measurement. Y.K. participated in early discussions and experiments.

D.E. and J.H. advised on the experiments and data analysis. All authors discussed the results and commented on the paper.

## Additional information

**Competing interests:** The authors declare no competing interests.

