## [Peer Review File · Nature Communications]

Reviewers' comments:

Reviewer #1 (Remarks to the Author):

This manuscript entitled "Thermal Radiation Control from Hot Graphene Electrons Coupled to a Photonic Crystal Nanocavity" reports a high-temperature thermal emitter with selective emission from a graphene-nanocavity. The authors show that the cavity strongly modifies the EM LDOS in the NIR spectrum, and the electron temperature of graphene is highly decoupled from lattice phonons, resulting in comparatively cool temperature (700 K) of the photonic crystal nanocavity. However, the novelty and the new insight of the thermal emitters are poor in the author's manuscript because authors don't cite the previous important reports on nanocarbon-based microcavity devices and thermal emitters, and don't discuss on the novelty, importance and new physics compared with their reports. In addition, the observed results can be explained by the simple models of microcavities with thermal emitters, which have been already reported. Furthermore, the reliability of the estimated values (e.g., graphene temperature and cavity temperature) is low. Therefore, I think that this manuscript is not suitable for Nature Communications.

1. Many papers on nanocarbon-based blackbody emitters have been reported, but the authors cite few relevant papers on graphene-based blackbody emitters.

[1] Freitag, M., Chiu, H.-Y., Steiner, M., Perebeinos, V. & Avouris, P. Thermal infrared emission from biased graphene. *Nat. Nanotechnol.* 5, 497-501 (2010).

[2] Bae, M.-H., Islam, S., Dorgan, V. E. & Pop, E. Scaling of high-field transport and localized heating in graphene transistors. *ACS Nano* 5, 7936-7944 (2011).

[3] Luxmoore, I. J., Adlem, C., Poole, T., Lawton, L. M., Mahlmeister, N. H. & Nash, G. R. Thermal emission from large area chemical vapor deposited graphene devices. *Appl. Phys. Lett.* 103, 131906 (2013).

[4] Lawton, L. M., Mahlmeister, N. H., Luxmoore, I. J. & Nash, G. R. Prospective for graphene based thermal mid-infrared light emitting devices. *AIP Adv.* 4, 087139 (2014).

2. In addition, it is unfair that the previous important reports on the nanocarbon-based light emitter with a "microcavity" (shown as follows) was not cited. The observed emission properties in Fig. 1, 2 4 in the manuscript can be explained by the simple microcavity model with thermal light emitters, which have been reported in the following reports. Especially, the explanation of emission spectrum with "local density of state" is essentially same model in the following reports, and there is no novel physics. Furthermore, the authors state the "high-temperature thermal emitter with selective emission from a graphene-nanocavity"; however, the following reports also showed the high-temperature (1100 and 1500K) nanocarbon thermal emission from a microcavity. There is no essential difference between this manuscript and these previous reports on it.

[5] Engel, M. et al. Light-matter interaction in a microcavity-controlled graphene transistor. *Nat. Commun.* 3, 906 (2012).

[6] Fujiwara, M., Tsuya, D. & Maki, H. Electrically driven, narrow-linewidth blackbody emission from carbon nanotube microcavity devices. *Appl. Phys. Lett.* 103, 143122 (2013).

[7] Pyatkov, F., Fütterling, V., Khasminskaya, S., Flavel, B., Hennrich F., Kappes, M., Krupke, & R., Pernice, W., Cavity-enhanced light emission from electrically driven carbon nanotubes. *Nat. Photonics* 10, 420-427 (2016).

3. The authors state that comparatively cool temperature (700 K) of the photonic crystal nanocavity (i.e., Si substrate) is realized because the electron bath in graphene is highly decoupled from lattice phonons. However, in the previous report on the graphene-based thermal emitters in the following Ref. [8], which is not cited in the manuscript, the temperature rise of the substrate is very low compared with the temperature rise of graphene under thermal emission, where the temperature rise of the substrate is $\sim 1/6$ of that of graphene. This significant temperature difference between graphene and substrate can be explained by relatively high

thermal contact resistance (i.e., low thermal conductance) between graphene and substrate. This indicates that the temperature difference between graphene and the photonic crystal can be explained by the simple heat conduction model, taking into account “graphene-BN” and “BN-photonic crystal” thermal contact resistance without consideration of decoupling of electron bath from acoustic photon bath.

[8] Miyoshi, Y., Fukazawa, Y., Amasaka, Y., Reckmann R., Yokoi T., Ishida, K., Kawahara, K., Ago, H., & Maki, H., High-speed and on-chip graphene blackbody emitters for optical communications by remote heat transfer. *Nat. Comm.* 9, 1279 (2018).

4. Furthermore, as shown in Ref [9] and [10], which are not cited in the manuscript, the temperature rise of the substrate is suppressed more for BN/graphene/BN thermal emitters, because of the increase of lateral heat conduction in BN. This also explain the large temperature difference between graphene and cavity without consideration of decoupling of electron bath from acoustic photon bath.

[9] Barnard, H. R., Zossimova, E., Mahlmeister, N. H., Lawton, L. M., Luxmoore, I. J. & Nash, G. R. Boron nitride encapsulated graphene infrared emitters. *Appl. Phys. Lett.* 108, 131110 (2016).

[10] Mahlmeister, N. H., Lawton, L. M., Luxmoore, I. J. & Nash, G. R. Modulation characteristics of graphene-based thermal emitters. *Appl. Phys. Express* 9, 012105 (2016).

5. The authors show a reference blackbody emission spectrum at 2050 K by a broken curve in Fig. 2(b). However, the spectral shape of this reference blackbody curve is entirely different from the ideal spectral shape of Planck’s law. In the wavelength range from 1450 to 1650 nm, the intensity of blackbody radiation should not be zero value and be almost flat at ~ 2000 K (e.g., see <https://physicsabout.com/black-body-radiation/>). In addition, the accuracy of the estimated electron temperature of graphene is very low because of the use of the unreliable spectra of blackbody radiation.

6. In Fig. 4(b), the authors estimate the cavity temperature from the wavelength shift of the resonant peaks. The authors state that the cavity temperature is given by the formulas of “ $\lambda(T) = \lambda(T_0) + \alpha(T-T_0)$ ” and “ $\alpha = 0.01$ nm/K”. However, the estimated cavity temperature in Fig. 4(c) (blue curve) is completely different from the calculated temperature by these formula (e.g., 40-nm shift corresponds to the temperature rise of ~ 4000 K).

7. On page 9, the authors state that the electron gas temperature far exceeds the cavity temperature due to the unusually low coupling between the graphene electron gas and graphene phonons. However, as mentioned above 3, the temperature difference between the graphene electron temperature and the cavity temperature can be simply understood by the thermal contact resistance between graphene and a cavity.

8. The authors confuse the graphene temperature with the cavity temperature. Especially, acoustic temperature of “graphene” is confused with the “cavity” temperature.

9. In the Si cavity, the thermo-optic coefficient (thermo-optic effect) is mainly dominated not by the graphene temperature but by the Si temperature. Hence, the resonant peak shift in Fig. 4(a) should be explained not by the graphene phonon temperature but by the Si phonon temperature. As mentioned above 3 and 9, Si temperature is dominated by the thermal contact resistance between graphene and Si, indicating that there is no novel physics in the demonstration of the thermo-optic effect in Fig. 4.

10. As shown in Ref. 39 in the manuscript and above Ref. [8], the direct quantum thermal coupling between graphene carriers and surface polar phonon of a substrate has been reported. To understand the detailed properties of thermal emission from a cavity, the consideration of this direct coupling between graphene and the substrate might be necessary.

11. The authors state that the thermal emitters can be applied for QKD in the main text of the manuscript. However, no relevant data is shown in this manuscript.

12. The high-speed thermal radiation is shown in Fig. 4(b). However, the high-speed thermal emission has been reported in Ref. [7], [8], [11], which are not cited in the manuscript.

[11] Mori, T., Yamauchi, Y., Honda, S. & Maki, H. An electrically driven, ultrahigh-speed, on-chip light emitter based on carbon nanotubes. *Nano Lett.* 14 3277–3283 (2014).

Reviewer #2 (Remarks to the Author):

As a first remark, I shall state that I believe the technological import of the device concept proposed in this work is indeed very significant. Features like the emission intensity, the wavelength range, and most importantly the high on-off switching speed, make it potentially a game changer for many applications. In fact, the authors concentrated on photonics applications and only briefly commented about optical sensors, but the latter would immensely benefit in terms of power consumption from a thermal IR source that could be reliably switched ON only for the short time needed to take a measurement. So, I have no doubt the use of graphene here is not merely scientifically interesting but does solve a real problem.

That said, however, I have a problem in gauging the impact of this paper in terms of absolute novelty. The concept of a graphene-based broadband thermal emitter has been already outlined by some of the authors in Ref 35, where measurements of the electronic temperature, of the lower lattice temperature of graphene and BN (Fig 3d), and fast response speed have been provided. Some results reported here are a repetition, or an obvious extension, of established ones. Other questions not mentioned in this paper such as long-term stability of the device and why BN encapsulation is crucial are also answered in Ref 35. The main message of this paper is the fabrication of a spectrally selective blackbody emitter. To cite the authors: "A variety of structures have been developed to tailor thermal radiation in this way, including optical gratings[6], photonic crystals[1, 7, 8], photonic cavities[9, 10], nano-antenna[11], and metamaterials[12]", so the idea of tailored blackbody emission is hardly a new one, the only challenge to overcome is the high operational temperature. But Ref 35 presents an emitter with sufficiently low lattice temperature, hence, this paper simply makes $1+1=2$ and does not seem to offer a real creative breakthrough. I emphasize that I appreciate that this paper does offer the necessary experimental demonstration of the concept through skilful fabrication and characterization, but this is very close to the definition of incremental work.

Still, because of the technological relevance of the results presented, I am inclined to suggest the editor should consider the opportunity of publishing this work in Nature Comms. To turn "inclined" into "convinced", there is one important aspect of the story I would like to see addressed. I could not see anywhere, neither in this manuscript nor in Ref 35, some clear evidence that the electron temperature reaches its maximum saturation value for short pulses. Also, a cavity temperature of $\sim 700\text{K}$ may still be too high for certain implementations. However, as mentioned above, operating this device under steady-state bias rather than in pulsed mode would invalidate one of its major and most useful strengths. Thus, a very important piece of information would be to present the electron temperature and the cavity temperature as a function of pulse width and duty cycle (at 13V or so), intended as the time the bias is applied versus the interval between consecutive pulses. I would not be surprised if under suitable pulse conditions T_e can still saturate to $\sim 2000\text{K}$ while the cavity T stays well below 700K . This would mark a much stronger step forward from Ref 35.

Reviewer #3 (Remarks to the Author):

Shiue et al. present an experimental investigation of a graphene based thermal emitter with an emission spectrum tailored via the integration of a silicon photonic crystal cavity. Through independent measurements of the electronic temperature in the graphene and the thermal properties of the PhC cavity, it is inferred that very high electronic temperatures required for efficient thermal emission can be achieved whilst the PhC remains relatively cool. In my opinion, the work is interesting and relatively novel; however, there are a number of omissions and points of clarification to be addressed.

1. On several occasions, justifications are made based on FDTD simulations that are not shown or referenced, including:
 - a. "This air-slot cavity increases the coupling rate with a 2D material on the PPC surface by almost a factor of three compared with a linear three-hole defect (L3) cavity".
 - b. "The Q factors dropped to 520, 430 and 400, respectively, because of the excess absorption of graphene to the cavity field. The degradation of Q due to the 25-nm-thick hBN layers is negligible, as tested in separate PPC cavities without graphene, consistent with simulation results."
 - c. "In our spectroscopy setup, the radiation of the cavity-graphene only couples to the microscope objective mode with a coupling efficiency $\eta \sim 0.1$, as calculated from 3D FDTD simulations."
2. The process for calibrating the thermal emission is not that clear. If I understand correctly, the temperature of the BB has been varied until the intensity matches the peak intensity of the PhC mode at $\sim 1600\text{nm}$, where the BB is scaled by 0.07 to take account for the reduced coupling efficiency to the objective. If this is correct, then it implies your combined device in the current set-up performs considerably worse than in the absence of a PhC in terms of light extraction efficiency. Is this due to the particular directionality of the PhC mode used? With a different cavity design, could you improve the directionality? Could you please comment?
3. In Fig. 1(e) you refer to three modes, but there are clearly at least five peaks that can be resolved. What happens to the shortest wavelength modes when the graphene is added?
4. Related to 3, in Fig. 2(b), as V_{ds} is increased, the emission spectrum changes considerably, with the relative intensity of modes changing, and additional peaks emerging. Can you explain this? The emission spectrum in Fig. 3(c) looks considerably different from the reflection spectrum in Fig 1(e).
5. Prior work on mid-IR hBN encapsulated graphene thermal emitters, tailored emission and ultrafast operation has not been recognised, including Barnard et al., Appl Phys Lett 2016, Shi et al., Nano Research 2018, and Miyoshi, Nature Comms 2018.

Response letter to reviewers

Reviewers' comments:

Reviewer #1 (Remarks to the Author):

This manuscript entitled “Thermal Radiation Control from Hot Graphene Electrons Coupled to a Photonic Crystal Nanocavity” reports a high-temperature thermal emitter with selective emission from a graphene-nanocavity. The authors show that the cavity strongly modifies the EM LDOS in the NIR spectrum, and the electron temperature of graphene is highly decoupled from lattice phonons, resulting in comparatively cool temperature (700 K) of the photonic crystal nanocavity. However, the novelty and the new insight of the thermal emitters are poor in the author’s manuscript because authors don’t cite the previous important reports on nanocarbon-based microcavity devices and thermal emitters, and don’t discuss on the novelty, importance and new physics compared with their reports. In addition, the observed results can be explained by the simple models of microcavities with thermal emitters, which have been already reported. Furthermore, the reliability of the estimated values (e.g., graphene temperature and cavity temperature) is low. Therefore, I think that this manuscript is not suitable for Nature Communications.

We appreciate the reviewer’s comments, which have allowed us to improve our manuscript. We have addressed all of the concerns one-by-one, as provided below.

The first major criticism is that we didn’t cite and compare to previous works on nanocarbon-based microcavities and thermal emitters. To address this concern, we now cite the references proposed by the reviewer, and we provide a brief discussion comparing the novelty of our work against the previous nanocarbon-based emitters. Very briefly, our work is the first that comprehensively studies high temperature (we show the highest electron gas temperature in a cavity ever reported); high-speed modulation (we show by far the highest modulation rate for a spectrally modified blackbody light source, to our knowledge); and the first independent temperature measurements of the cavity substrate vs. the blackbody electron gas (a capability uniquely enabled by our cavity-coupled approach and never demonstrated below). The physical properties of graphene differ from carbon nanotube in several respects that are of great importance for thermal emission, including:

1. The ability to laterally pattern the material and -- in this study -- align the hot-spot of the graphene device exactly with the cavity mode maximum for optimal coupling of the hot electron gas to the nanocavity;
2. Low-loss Ohmic contacts to the contacts, thanks to a much larger contact area than for a carbon nanotube: this is very important for efficient heating of the electron gas near the cavity
3. The availability of reliable encapsulation methods of graphene by hBN, which is critical for electron scattering reduction to achieve high saturation current and passivation for device durability.

We note that we did cite previous work on carbon nanotube thermal emitters. However, upon reflecting on the reviewer’s helpful comments, we recognize that a more substantive treatment of these early works should be presented. We hope that the revised sections, which are copied below with modifications highlighted in red, will achieve this.

- Nanocarbon-based materials including graphene[14-18] and carbon nanotubes[19-21] have emerged as promising thermal emitters alternative to semiconductor-based materials. Previous demonstrations show that they can support high saturation current density[14,17], ultrafast heating (cooling) modulation[15,18,21], and flexible integration with existing electronic and photonic technology[16,19,20].

- We further etched the hBN/graphene/hBN stack into a bowtie shape and aligned the central narrow strip to the PPC cavity area. This bowtie-shaped graphene facilitates a heated hot-electron spot in the graphene device to achieve optimal coupling of the hot-electron radiation to the cavity resonant field.
- We briefly comment on the key features that lead to the high modulation speed and the exceptionally high temperature of the thermal emitter. (1) The bowtie shape of the hBN-encapsulated graphene allows the precise spatial coupling of the hot electron gas at its narrowest section with the nanocavity; future work could further improve the mode overlap by a smaller (< 100 nm) taper of the graphene layer coupled to slot-mode nanocavities[26,45]. (2) The fan-out of the graphene layer to the metal contacts produces relatively low contact resistance, approximated to be 65 Ohm; this resistance may be further reduced by graphite/graphene contact with optimal orientation[46]. The S11 in Fig. S7(b) indicates a RC-limited response time ~ 0.19 ns. This response time is also similar to the expected acoustic phonon cooling time ~ 0.2 ns. (3) The high electron temperature reaching up to 2000K was enabled in part by the weak coupling between graphene electrons and acoustic phonons and the hBN encapsulation; for comparison, previous demonstration with CNTs and graphene without hBN encapsulation reported 1500K[20] and 1100 K[16], respectively. A temperature increase from 1500 K to 2000K translates to a three-fold higher thermal radiation intensity since the thermal radiated power scales as T^4 (Stefan-Boltzmann law). Even higher electron gas temperatures up to 2800 K, which were reported for suspended graphene monolayers, could further improve the emitter's radiative efficiency.

1. Many papers on nanocarbon-based blackbody emitters have been reported, but the authors cite few relevant papers on graphene-based blackbody emitters.

[1] Freitag, M., Chiu, H.-Y., Steiner, M., Perebeinos, V. & Avouris, P. Thermal infrared emission from biased graphene. *Nat. Nanotechnol.* 5, 497-501 (2010).

[2] Bae, M.-H., Islam, S., Dorgan, V. E. & Pop, E. Scaling of high-field transport and localized heating in graphene transistors. *ACS Nano* 5, 7936-7944 (2011).

[3] Luxmoore, I. J., Adlem, C., Poole, T., Lawton, L. M., Mahlmeister, N. H. & Nash, G. R. Thermal emission from large area chemical vapor deposited graphene devices. *Appl. Phys. Lett.* 103, 131906 (2013).

[4] Lawton, L. M., Mahlmeister, N. H., Luxmoore, I. J. & Nash, G. R. Prospective for graphene based thermal mid-infrared light emitting devices. *AIP Adv.* 4, 087139 (2014).

As detailed in the paragraph above, we addressed this question by adding more discussion on previously demonstrated thermal emitters based on nanocarbon materials with corresponding citations.

2. In addition, it is unfair that the previous important reports on the nanocarbon-based light emitter with a "microcavity" (shown as follows) was not cited. The observed emission properties in Fig. 1, 2 4 in the manuscript can be explained by the simple microcavity model with thermal light emitters, which have been reported in the following reports. Especially, the explanation of emission spectrum with "local density of state" is essentially same model in the following reports, and there is no novel physics. Furthermore, the authors state the "high-temperature thermal emitter with selective emission from a graphene-nanocavity"; however, the following reports also showed the high-temperature (1100

and 1500K) nanocarbon thermal emission from a microcavity. There is no essential difference between this manuscript and these previous reports on it.

[5] Engel, M. et al. Light–matter interaction in a microcavity-controlled graphene transistor. *Nat. Commun.* 3, 906 (2012).

[6] Fujiwara, M., Tsuya, D. & Maki, H. Electrically driven, narrow-linewidth blackbody emission from carbon nanotube microcavity devices. *Appl. Phys. Lett.* 103, 143122 (2013).

[7] Pyatkov, F., Fütterling, V., Khasminskaya, S., Flavel, B., Hennrich F., Kappes, M., Krupke, & R., Pernice, W., Cavity-enhanced light emission from electrically driven carbon nanotubes. *Nat. Photonics* 10, 420-427 (2016).

We disagree strongly that “there is no essential difference between the manuscript and previous reports.” The physical properties of graphene are very distinct and -- in many cases advantageous -- over carbon nanotubes previously studied. They contribute to our demonstrations of record modulation rate and record temperature (and corresponding thermal radiation intensity). For this reason, we believe that our manuscript holds much novelty. In particular, as noted above: the ability to pattern the material and -- in this study -- align the hot-spot of the graphene device exactly with the cavity mode maximum, thus allowing engineering of the coupling of the hot-electron gas to the cavity; low-loss Ohmic contacts to the contacts, thanks to a much larger contact area than for a carbon nanotube.

The physical model in our work is also distinct from previous reports on CNT-coupled thermal emitters, since we account explicitly for the electron gas temperature, which is uniquely decoupled from the atomic lattice in graphene. We emphasize that this electron-phonon decoupling is especially important as it allows far higher temperatures -- in our case up to 2000 K vs 1500 K for CNTs (1100 K for graphene without hBN encapsulation). Since thermal radiated power scales as T^4 , this difference amounts to a three-fold increase in the thermal radiative power.

Furthermore, the temperature of graphene electron gas can be Ohmically heated even higher -- a recent experimental work reached 2800C (though not coupled to a cavity), which indicates advantage in photon flux of $(2800/1500)^4 \sim 12$. Moreover, the total radiation power of graphene-based devices can be scaled up because graphene is 2D rather than the 1D geometry of the CNT. The filling factor of graphene to the cavity mode area can be easily engineered to achieve optimal efficiency and output intensity at large scale.

We also studied, for the first time, the hBN-encapsulated graphene in a cavity. Optical phonon (OP) energy of hBN (~ 150 meV) is much higher than SiO_2 [1-6,8] and Al_2O_3 [8], which are ~ 60 meV and 20 meV, respectively. Higher OP energy means less scattering of hot graphene electrons with dielectric, enabling us to achieve the highest temperature among nanocarbon-based emitters in a cavity. Among previous reports, including Ref. 5-7, our device is also the only demonstration of a spectrally-modified high temperature emitter with high-speed modulation. The implication of such a high-speed emitter is crucial to impact a broad scope of sensing, spectroscopy and other photonic applications.

3. The authors state that comparatively cool temperature (700 K) of the photonic crystal nanocavity (i.e., Si substrate) is realized because the electron bath in graphene is highly decoupled from lattice phonons. However, in the previous report on the graphene-based thermal emitters in the following Ref. [8], which is not cited in the manuscript, the temperature rise of the substrate is very low compared with the temperature rise of graphene under thermal emission, where the temperature rise of the substrate is $\sim 1/6$ of that of graphene. This significant temperature difference between graphene

and substrate can be explained by relatively high thermal contact resistance (i.e., low thermal conductance) between graphene and substrate. This indicates that the temperature difference between graphene and the photonic crystal can be explained by the simple heat conduction model, taking into account “graphene-BN” and “BN-photonic crystal” thermal contact resistance without consideration of decoupling of electron bath from acoustic photon bath.

[8] Miyoshi, Y., Fukazawa, Y., Amasaka, Y., Reckmann R., Yokoi T., Ishida, K., Kawahara, K., Ago, H., & Maki, H., High-speed and on-chip graphene blackbody emitters for optical communications by remote heat transfer. Nat. Comm. 9, 1279 (2018).

The most important consequence of the weak coupling between the electron gas and the graphene phonons is that a high electron gas temperature can be achieved for a given electrical power. This decoupling is particularly strong in graphene, which partially explains why such high electron gas temperatures are achievable. Therefore, it is critical that our physical model include a quantitative description of the electron gas temperature. The reviewer notes above that the substrate temperature in Ref.[8] is only elevated by ~% of that of the graphene. However, Ref.[8] does not experimentally measure this temperature; in our work, we can uniquely take advantage of the suspended photonic crystal temperature as a direct thermometer of the substrate temperature. We also emphasize that maintaining a low substrate temperature: we instead mean that the *suspended photonic crystal cavity* is maintained at relatively low temperature, which is important because other works based on heated photonic crystal devices are ultimately limited by the material stability of that cavity medium. The ability to maintain an electron gas thermal emitter at much higher temperature than the photonic crystal cavity is key to our demonstration.

To avoid possible misunderstanding, we modified the following sections (see red highlights):

- “A striking conclusion is that the electron gas temperature of graphene (Fig. 3(c)) far exceeds the suspended silicon PPC cavity temperature (Fig. 4(b)): the electron gas can be hot – as desired for thermal radiator – while the nanophotonic substrate, which strongly modifies the emission spectrum through the EM LDOS, remains comparatively cool.”

We also compare the electro-thermal models among Ref. 8, 9, 10 and our work. The temperature of the substrates in previous studies can be kept at ~ 300 K as the electron temperature reaches ~ 700 K due to the thermal resistance between graphene and the substrate. In this moderate temperature regime, the thermal resistance between graphene and the substrate is more dominant than the effects of temperature decoupling between graphene electron and acoustic phonons (AP). The electron-AP decoupling could be negligible if the interface thermal resistance is high.

However, as the electron temperature and electric field continue to increase at higher voltage, the saturation velocity of the electron dramatically decreases with voltage increment. The resulting saturated current-voltage (I-V) characteristic serves as an important indicator to gauge two facts: (1) the saturation current corresponds to the saturation velocity and the electron temperature of graphene, and (2) the transition of the I-V curve (i.e. dI/dV) relates to the electron-AP temperature decoupling in graphene qualitatively. Our measured I-V curve agrees well with the Joule heating model considering decoupled electron and AP temperatures. Without electron-AP decoupling at high temperature and high electric field, the I-V curve would become a complete flat line (i.e. constant current) for voltage > 5 V, which conflicts with experimentally measured results.

The decoupling of electron-AP temperature has been widely studied and verified in Ref. 14-15, 25 of the manuscript and Ref. 1-6 in the supporting information when electron temperature goes beyond 700 K. As reported in the literature, this decoupling is a direct result of the AP cooling bottleneck of graphene and carbon nanotubes at high electron temperature, which should also be taken into account in our electro-thermal modeling. Thus, we believe it is critical to include thermal bath decoupling between acoustic phonons and graphene electrons to capture the operation and physics of our device.

4. Furthermore, as shown in Ref [9] and [10], which are not cited in the manuscript, the temperature rise of the substrate is suppressed more for BN/graphene/BN thermal emitters, because of the increase of lateral heat conduction in BN. This also explain the large temperature difference between graphene and cavity without consideration of decoupling of electron bath from acoustic photon bath.

[9] Barnard, H. R., Zossimova, E., Mahlmeister, N. H., Lawton, L. M., Luxmoore, I. J. & Nash, G. R. Boron nitride encapsulated graphene infrared emitters. *Appl. Phys. Lett.* 108, 131110 (2016).

[10] Mahlmeister, N. H., Lawton, L. M., Luxmoore, I. J. & Nash, G. R. Modulation characteristics of graphene-based thermal emitters. *Appl. Phys. Express* 9, 012105 (2016).

As stated in the answer to Q3, our measured I-V characteristics conflict with the assumption that electron and AP are at the same temperature in a regime of high electron temperature (> 700 K) and high bias voltage (> 5 V).

5. The authors show a reference blackbody emission spectrum at 2050 K by a broken curve in Fig. 2(b). However, the spectral shape of this reference blackbody curve is entirely different from the ideal spectral shape of Planck's law. In the wavelength range from 1450 to 1650 nm, the intensity of blackbody radiation should not be zero value and be almost flat at ~ 2000 K (e.g., see <https://physicsabout.com/black-body-radiation/>). In addition, the accuracy of the estimated electron temperature of graphene is very low because of the use of the unreliable spectra of blackbody radiation.

We apologize for any confusion that our presentation of the data and model may have caused. The dashed line in Figure 2(b) was intended to represent the expected collection of a reference (unpatterned) blackbody radiator after passing through our collection and detection apparatus. The modulation in the spectrum, and the eventual cut-off around 1620 nm, is due to optical component transmission and due to our InGaAs camera cutoff at the long-wavelength tail. We had included this curve to allow a calibration of the graphene-hBN-cavity blackbody radiation as collected in our apparatus. However, after reading the comments by reviewers, we now realize that this presentation has led to some confusion. We therefore now added Figure 2(c) that plots the spectral radiance in units of $\text{kW}/\text{sr}/\text{m}^2/\text{nm}$, as a function of wavelength. The original curve is now moved to the apparatus calibration section in supporting information (SI). We also revised the calibration section in SI accordingly.

6. In Fig. 4(b), the authors estimate the cavity temperature from the wavelength shift of the resonant peaks. The authors state that the cavity temperature is given by the formulas of " $\lambda(T) = \lambda(T_0) + \alpha(T-T_0)$ " and " $\alpha = 0.01 \text{ nm}/\text{K}$ ". However, the estimated cavity temperature in Fig. 4(c) (blue curve) is

completely different from the calculated temperature by these formula (e.g., 40-nm shift corresponds to the temperature rise of ~4000 K).

We are grateful to the reviewer for catching this typographical error. $\alpha = 0.1 \text{ nm/K}$, or $\alpha = 100 \text{ pm/K}$. Thus, an increment of 400K results in a ~40 nm wavelength shift, bringing the cavity temperature to ~700K. This is now fixed in the manuscript. 0.1 nm/K so 400K increment + 273 K room T , ~ 700 K

7. On page 9, the authors state that the electron gas temperature far exceeds the cavity temperature due to the unusually low coupling between the graphene electron gas and graphene phonons. However, as mentioned above 3, the temperature difference between the graphene electron temperature and the cavity temperature can be simply understood by the thermal contact resistance between graphene and a cavity.

There are four main considerations concerning the temperature difference between the electron gas temperature and the cavity: (1) the coupling rate between the electron gas and graphene phonons; (2) the coupling between graphene phonons with hBN and the PhC cavity; (3) the coupling rate between the Si cavity and the environment; and (4) the radiative cooling rate of the graphene electron gas to the environment. We do not disagree that (2) is important, but (1) is also essential for capturing our device. This is borne out by our model and the corresponding measurements. See above for longer discussion.

8. The authors confuse the graphene temperature with the cavity temperature. Especially, acoustic temperature of “graphene” is confused with the “cavity” temperature.

We do distinguish between the two different temperatures: Fig. 1c describes the heat dissipation paths with statements “... the heated electrons first reach equilibrium with optical phonons in graphene and hBN, coupling more slowly to the acoustic phonon bath by thermal conductance γ_e . The heat eventually dissipates to the substrate (via γ_0), which can remain at a much lower temperature than ...” in the corresponding paragraph.

However, we see now that we can be clearer in this discussion. Thus, prompted by the reviewer, we made the following changes to the manuscript:

- “... the thermal emission that arises from the graphene electron gas, whose temperature is highly decoupled from graphene's atomic lattice, can exceed 2000 K, while the surrounding Si cavity itself stays at only 700 K.”
- “A striking conclusion is that the electron gas temperature of graphene (Fig. 3(c)) far exceeds the suspended silicon PPC cavity temperature (Fig. 4(b)): the electron gas can be hot – as desired for thermal radiator – while the nanophotonic substrate, which strongly modifies the emission spectrum through the EM LDOS, remains comparatively cool.”

We thank the reviewer for pointing out this possible source of confusion.

9. In the Si cavity, the thermo-optic coefficient (thermo-optic effect) is mainly dominated not by the graphene temperature but by the Si temperature. Hence, the resonant peak shift in Fig. 4(a) should be explained not by the graphene phonon temperature but by the Si phonon temperature. As mentioned above 3 and 9, Si temperature is dominated by the thermal contact resistance between graphene and Si, indicating that there is no novel physics in the demonstration of the thermo-optic effect in Fig. 4.

We do not claim any novelty whatsoever about thermo-optic effects in silicon. But, we do *make use of the thermo-optic effect* as a way of directly measuring the temperature of the silicon PhC membrane near the cavity.

The reviewer states that “the resonant peak shift in Fig. 4(a) should be explained not by the graphene phonon temperature but by the Si phonon temperature.” We certainly agree with that statement, but we do not say anything to the contrary in the manuscript.

10. As shown in Ref. 39 in the manuscript and above Ref. [8], the direct quantum thermal coupling between graphene carriers and surface polar phonon of a substrate has been reported. To understand the detailed properties of thermal emission from a cavity, the consideration of this direct coupling between graphene and the substrate might be necessary.

Direct quantum thermal coupling between graphene electrons and surface polar phonons has been widely studied for graphene deposited on polar substrates. For graphene on silicon oxide, as reported in Ref. 8, 12, 13, surface phonons coupling plays an important role for electron cooling. In our device, graphene is in direct contact with hBN but not oxide. The in-plane heat conduction of hBN has much higher heat conductivity than the oxide remote phonon conduction, as also reported in Ref. 12. In fact, remote phonon coupling strength scales inversely with respect to the distance between graphene and the polar substrate, as studied in Ref. 14. Considering a 15-nm separation of graphene to the substrate and high heat conduction in hBN, the effects of surface phonon cooling of the substrate would be negligible.

The hot electrons in graphene do, however, couple to the surface phonon of hBN through direct quantum coupling. Since the geometries of graphene and hBN are identical, remote phonon coupling impacts the electro-thermal modeling the same way as heat conduction of hBN, which is already considered together in our model. To separate the effects of hBN heat conduction and remote phono-coupled heat conduction would require further experiment studies, which is outside the scope of this work.

11. The authors state that the thermal emitters can be applied for QKD in the main text of the manuscript. However, no relevant data is shown in this manuscript.

We thank the reviewer for pointing out this lack of clarity. We add a section in the supporting information to clarify this calculation. The added section is copied below

6. Incoherent photon rates for QKD

For a thermal emitter that couples to an one-dimensional (1-D) waveguide, the total emission power is limited by 1-D LDOS[22], giving

$$P_{\lambda}(T) = \frac{hc^2}{\lambda^3(e^{\frac{hc}{\lambda k_B T}} - 1)}$$

The total emission power then yields $I(T) = \eta \int_{\lambda}^{\lambda+\Delta\lambda} P_{\lambda}(T)d\lambda$, and $T = 2000$ K, $\lambda = 1.5 \mu m$, $-\Delta\lambda/\lambda = Q = 500$, and $\eta = 0.9$ is the coupling efficiency from the cavity to an on-chip waveguide.

The average emission photon number for a single pulse equals $n = \frac{I(T)\delta t}{hc/\lambda}$, and $\delta t = 100$ ps, giving $n = 0.2$ photons/pulse.

12. The high-speed thermal radiation is shown in Fig. 4(b). However, the high-speed thermal emission has been reported in Ref. [7], [8], [11], which are not cited in the manuscript.

As addressed in the answer to Q1, we have added corresponding citations in the modified paragraph to discuss previous achievements in nanocarbon-based emitters.

[11] Mori, T., Yamauchi, Y., Honda, S. & Maki, H. An electrically driven, ultrahigh-speed, on-chip light emitter based on carbon nanotubes. *Nano Lett.* 14 3277–3283 (2014).

[12] Li, X., Kong, J.M. Zavada, and Kim, K. W. Strong substrate effects of Joule heating in graphene electronics. *Appl. Phys. Lett.* 99, 233114 (2011)

[13] Tee Kan Koh, Austin S. Lyons, Myung-Ho Bae, Bin Huang, Vincent E. Dorgan, David G. Cahill, and Eric Pop, Role of remote interfacial phonon (RIP) scattering in heat transport across graphene/SiO₂ interfaces. *Nano Lett.* 16, 6014-6020 (2016).

[14] Fratini, S., Guinea, F. Substrate-limited electron dynamics in graphene. *Phys. Rev. B* 77, 195415 (2008)

Reviewer #2 (Remarks to the Author):

As a first remark, I shall state that I believe the technological import of the device concept proposed in this work is indeed very significant. Features like the emission intensity, the wavelength range, and most importantly the high on-off switching speed, make it potentially a game changer for many applications. In fact, the authors concentrated on photonics applications and only briefly commented about optical sensors, but the latter would immensely benefit in terms of power consumption from a thermal IR source that could be reliably switched ON only for the short time needed to take a measurement. So, I have no doubt the use of graphene here is not merely scientifically interesting but does solve a real problem.

That said, however, I have a problem in gauging the impact of this paper in terms of absolute novelty. The concept of a graphene-based broadband thermal emitter has been already outlined by some of the authors in Ref 35, where measurements of the electronic temperature, of the lower lattice

temperature of graphene and BN (Fig 3d), and fast response speed have been provided. Some results reported here are a repetition, or an obvious extension, of established ones. Other questions not mentioned in this paper such as long-term stability of the device and why BN encapsulation is crucial are also answered in Ref 35. The main message of this paper is the fabrication of a spectrally selective blackbody emitter. To cite the authors: "A variety of structures have been developed to tailor thermal radiation in this way, including optical gratings[6], photonic crystals[1, 7, 8], photonic cavities[9, 10], nano-antenna[11], and metamaterials[12]", so the idea of tailored blackbody emission is hardly a new one, the only challenge to overcome is the high operational temperature. But Ref 35 presents an emitter with sufficiently low lattice temperature, hence, this paper simply makes 1+1=2 and does not seem to offer a real creative breakthrough. I emphasize that I appreciate that this paper does offer the necessary experimental demonstration of the concept through skilful fabrication and characterization, but this is very close to the definition of incremental work.

Still, because of the technological relevance of the results presented, I am inclined to suggest the editor should consider the opportunity of publishing this work in Nature Comms. To turn "inclined" into "convinced", there is one important aspect of the story I would like to see addressed. I could not see anywhere, neither in this manuscript nor in Ref 35, some clear evidence that the electron temperature reaches its maximum saturation value for short pulses. Also, a cavity temperature of ~700K may still be too high for certain implementations. However, as mentioned above, operating this device under steady-state bias rather than in pulsed mode would invalidate one of its major and most useful strengths. Thus, a very important piece of information would be to present the electron temperature and the cavity temperature as a function of pulse width and duty cycle (at 13V or so), intended as the time the bias is applied versus the interval between consecutive pulses. I would not be surprised if under suitable pulse conditions T_e can still saturate to ~2000K while the cavity T stays well below 700K. This would mark a much stronger step forward from Ref 35.

We thank the reviewer for his positive feedback regarding the technology breakthrough presented in this paper. To further gain insights of the transient temperature of the graphene emitter and the substrate temperature elevation at high-speed operation, we added two studies: (1) experimentally measured thermal radiation intensity with respect to different electrical pulse durations; (2) transient temperature simulations of the graphene hot electrons based on experimentally measured thermal relaxation time constants.

For part (1), we have shown that as the excitation pulse duration increases, the electron temperature of graphene starts to saturate, reaching a maximum of 1550K with 10 V peak voltage. The extracted rising (falling) time constant for the electron temperature is around 0.18 ns, which could be attributed to the combined effect of acoustic phonon cooling rate and resistor-capacitor response of the graphene device. For part (2), we applied experimentally measured cooling (heating) time constants to electron-phonon scattering equations and solved for the transient temperature of graphene electrons, phonons, and the silicon substrate. The results shows that under 0.5-ns-long electrical pulsed excitation (duty cycle of 10%), the electrons in graphene can reach a temperature of 1950 K while the substrate temperature remains around 360 K. These results show that our cavity-graphene emitter can reach high electron temperature with short electrical pulses. In this regime, the emitter can be even more efficient than being operated at DC since the substrate temperature elevation is greatly suppressed.

The following table shows the added or modified parts in the manuscript and SI to incorporate studies (1) and (2).

Location	Content	Revision
Manuscript	As shown in Fig. 4(c), we investigate the temporal response of the graphene emission with an 100-ps electrical pulse excitation (Supporting Info). The temporal response of the thermal emission shows a full width at half maximum (FWHM) of 350 ps, indicating an on-off modulation speed > 1 GHz. Varying the excitation pulse duration ΔT from 0.1 ns to 2 ns, we observed that the emission intensity started to saturate for $\Delta T > 1$ ns, corresponding to a saturation temperature of 1550 K, as shown in Fig. S4. We numerically simulated the transient temperature of the graphene emitter, showing that the substrate temperature elevation can be reduced to only 50 K with 10%-duty-cycle electrical pulses (Supporting Info). As we demonstrated in separate work[15], an optimized graphene device allows an on-off modulation speed of the thermal emission at a rate exceeding 10 GHz – comparable to fast gain-switched lasers.	Added red descriptions.
SI Sec. 3	Using TCSPC method, The agreement of the experiment to the theory curves indicates that the electron temperature of graphene starts to saturates at 1550 K when $T > 1$ ns, i.e. it is possible to drive the electron temperature of graphene to its steady-state temperature with short excitation durations. Here the time constant τ of 0.18 ns may be limited by the acoustic phonon cooling or the electrical resistor-capacitor response of the graphene device.	Added 4 paragraphs
SI Sec. 4	4. Thermal Relaxation Time Constant and Transient Response of the Cavity-Graphene Emitter As shown in Fig. 1(c), the dominant cooling pathway of the graphene hot electrons consists of initial quasi-equilibrium of hot graphene electrons and optical phonons, followed by subsequent cooling of acoustic phonon at a rate of γ_e, and to the substrate at a rate of γ_0, and finally return to the ambient. The time-dependent energy relaxation follows equations ... Fig. S5(d) shows the simulated T_e, T_{ap} and T_{sub} with 500-ps-long electrical pulses repeated every 5 ns (200 MHz, DC=10%). It is seen that the substrate temperature were elevated by only 50 K because of such a short duration of the graphene heating. The electron temperature, however, still reaches around 1950 K due to the fast thermal response and energy relaxation bottleneck to the acoustic phonons.	Add entire section

Reviewer #3 (Remarks to the Author):

Shiue et al. present an experimental investigation of a graphene based thermal emitter with an emission spectrum tailored via the integration of a silicon photonic crystal cavity. Through independent measurements of the electronic temperature in the graphene and the thermal properties of the PhC cavity, it is inferred that very high electronic temperatures required for efficient thermal emission can be achieved whilst the PhC remains relatively cool. In my opinion, the work is interesting and relatively novel; however, there are a number of omissions and points of clarification to be addressed.

1. On several occasions, justifications are made based on FDTD simulations that are not shown or referenced, including:

- a. "This air-slot cavity increases the coupling rate with a 2D material on the PPC surface by almost a factor of three compared with a linear three-hole defect (L3) cavity".
- b. "The Q factors dropped to 520, 430 and 400, respectively, because of the excess absorption of graphene to the cavity field. The degradation of Q due to the 25-nm-thick hBN layers is negligible, as tested in separate PPC cavities without graphene, consistent with simulation results."
- c. "In our spectroscopy setup, the radiation of the cavity-graphene only couples to the microscope objective mode with a coupling efficiency $\eta \sim 0.1$, as calculated from 3D FDTD simulations."

We thank the reviewer for noting this omission. We add section 5 in the supporting information to discuss the detailed properties of the air-slot cavity. This section addresses its mode profile, far-field coupling efficiencies and the effect of hBN on top of the PPC cavity.

2. The process for calibrating the thermal emission is not that clear. If I understand correctly, the temperature of the BB has been varied until the intensity matches the peak intensity of the PhC mode at $\sim 1600\text{nm}$, where the BB is scaled by 0.07 to take account for the reduced coupling efficiency to the objective. If this is correct, then it implies your combined device in the current set-up performs considerably worse than in the absence of a PhC in terms of light extraction efficiency. Is this due to the particular directionality of the PhC mode used? With a different cavity design, could you improve the directionality? Could you please comment?

The calibration process is performed exactly as the reviewer summarized. We have also revised section 2 of supporting info to make it more clearer. Regarding the coupling efficiencies, the emissivity of single layer graphene is about 2.3%, meaning the emission intensity of graphene will be 0.023 of ideal blackbody outside of a cavity. The total emission in our demonstrate is enhanced to 0.07, which is limited by the coupling efficiency of the PPC cavity. It is possible to introduce small perturbation to the original PPC lattice, resulting in $> 30\%$ coupling efficiency of the cavity emission in the vertical direction. We have included this discussion in the supporting info.

3. In Fig. 1(e) you refer to three modes, but there are clearly at least five peaks that can be resolved. What happens to the shortest wavelength modes when the graphene is added?

Due to the inherently lossy configuration of a cross-polarized microscope setup, the cavity peak may not be resolvable by cross-polarized reflection measurement if the out-coupling power falls below the noise background. Before graphene is deposited, the cavity modes generally exhibit high Q and high out-coupling power, therefore showing five pronounced peaks in the cross-polarized measurement. After graphene deposition, the cavity Q degrades dramatically, some modes, particularly for the shortest wavelength modes, become too lossy and can not be resolved by the cross-polarized reflection measurement.

4. Related to 3, in Fig. 2(b), as V_{ds} is increased, the emission spectrum changes considerably, with the relative intensity of modes changing, and additional peaks emerging. Can you explain this? The emission spectrum in Fig. 3(c) looks considerably different from the reflection spectrum in Fig 1(e).

Unlike reflection measurement in a cross-polarized setup, the emission measurement can have much higher signal-to-noise ratio (SNR) because there is very little background signal exists in the measured emission light. Owing to the improved SNR, most modes now can be resolved in the emission measurement, and the spectrum is qualitatively closer to that before graphene is deposit, showing four distinct peaks in the spectrum at $V_{DS} = 13$ V.

5. Prior work on mid-IR hBN encapsulated graphene thermal emitters, tailored emission and ultrafast operation has not been recognised, including Barnard et al., Appl Phys Lett 2016, Shi et al., Nano Research 2018, and Miyoshi, Nature Comms 2018.

We recognize that some references were not properly cited, particular for previously reported high-speed emitters based on nanocarbon materials. We have added new discussions along with the citations in the revised manuscript to address this issue. We thank the reviewer for pointing out this issue.

REVIEWERS' COMMENTS:

Reviewer #1 (Remarks to the Author):

I believe that the authors have carefully addressed the referee's concerns. I recommend the manuscript for publication.

Reviewer #2 (Remarks to the Author):

It appears that the novelty of the concept proposed was questioned also by other referees. Concerning carbon nanomaterials in general, it is true that a lack of referencing to previous work on carbon nanotube emitters had to be addressed, yet I agree with the authors that graphene offers several unique features as the material of choice and cannot be merely considered incremental work in this regard. My personal doubts related to graphene prior art specifically, as the basic ingredients of this paper appeared to have been already available (including from the authors themselves).

Upon revision of the resubmitted manuscript, I believe there is now sufficient novel import to grant publication of this paper. The idea may not be novel in itself, but the new experimental evidence provided (and related calculations) does represent a step forward towards the understanding and implementation of such devices. Regarding my own comments in particular, I was pleased to see how pulsed operation could allow to reduce substrate heating to fairly insignificant values. Of course, this aims at the possibility of fabricating these devices onto some integrated driving electronics, with thermal budgets typically much lower than the previously quoted 700K.

Reviewer #3 (Remarks to the Author):

I am satisfied that the comments in my original review have been addressed, with the exception of point 5, "Prior work on mid-IR hBN encapsulated graphene thermal emitters, tailored emission and ultrafast operation has not been recognised, including Barnard et al., Appl Phys Lett 2016, Shi et al., Nano Research 2018, and Miyoshi, Nature Comms 2018." Although this comment has been addressed in part, the work of Barnard was the first to report the stability of Boron nitride encapsulated graphene emitters, and this important contribution has not been properly acknowledged.

Response letter to the reviewers

REVIEWERS' COMMENTS:

Reviewer #1 (Remarks to the Author):

I believe that the authors have carefully addressed the referee's concerns. I recommend the manuscript for publication.

We thank the reviewer's positive feedback.

Reviewer #2 (Remarks to the Author):

It appears that the novelty of the concept proposed was questioned also by other referees. Concerning carbon nanomaterials in general, it is true that a lack of referencing to previous work on carbon nanotube emitters had to be addressed, yet I agree with the authors that graphene offers several unique features as the material of choice and cannot be merely considered incremental work in this regard. My personal doubts related to graphene prior art specifically, as the basic ingredients of this paper appeared to have been already available (including from the authors themselves).

Upon revision of the resubmitted manuscript, I believe there is now sufficient novel import to grant publication of this paper. The idea may not be novel in itself, but the new experimental evidence provided (and related calculations) does represent a step forward towards the understanding and implementation of such devices. Regarding my own comments in particular, I was pleased to see how pulsed operation could allow to reduce substrate heating to fairly insignificant values. Of course, this aims at the possibility of fabricating these devices onto some integrated driving electronics, with thermal budgets typically much lower than the previously quoted 700K.

We thank the reviewer's positive feedback and appreciate the point about pulsed driving in particular.

Reviewer #3 (Remarks to the Author):

I am satisfied that the comments in my original review have been addressed, with the exception of point 5, "Prior work on mid-IR hBN encapsulated graphene thermal emitters, tailored emission and ultrafast operation has not been recognised, including Barnard et al., Appl Phys Lett 2016, Shi et al., Nano Research 2018, and Miyoshi, Nature Comms 2018." Although this comment has been addressed in part, the work of Barnard was the first to

report the stability of Boron nitride encapsulated graphene emitters, and this important contribution has not been properly acknowledged.

We thank the reviewer's positive feedback. We also agree that the paper by Barnard et al., should be acknowledged for their achievement. We have cited this paper in the revised manuscript. The revised parts are attached below.

"Nanocarbon-based materials including graphene[14-19] and carbon nanotubes[20-22], have emerged as intriguing thermal emitters alternative to semiconductor-based materials."

"The high electron temperature reaching up to 2000 K was enabled in part by the weak coupling between graphene electrons and acoustic phonons and the hBN encapsulation[14, 15, 19]."

[19] Barnard, H. R. et al. Boron nitride encapsulated graphene infrared emitters. Applied Physics Letters 108, 131110 (2016).